# Human-ignited fires result in more extreme fire behavior and ecosystem impacts

Stijn Hantson [1,2✉], Niels Andela [3,4], Michael L. Goulden[5] & James T. Randerson [5]

California has experienced a rapid increase in burned area over the past several decades. Although fire behavior is known to be closely tied to ecosystem impacts, most analysis of changing fire regimes has focused solely on area burned. Here we present a standardized database of wildfire behavior, including daily fire rate-of-spread and fire radiative power for large, multiday wildfires in California during 2012–2018 using remotely-sensed active fire observations. We observe that human-ignited fires start at locations with lower tree cover and during periods with more extreme fire weather. These characteristics contribute to more explosive growth in the first few days following ignition for human-caused fires as compared to lightning-caused fires. The faster fire spread, in turn, yields a larger ecosystem impact, with tree mortality more than three times higher for fast-moving fires (>1 km day$^{-1}$) than for slow moving fires (<0.5 km day$^{-1}$). Our analysis shows how human-caused fires can amplify ecosystem impacts and highlights the importance of limiting human-caused fires during period of extreme fire weather for meeting forest conservation targets under scenarios of future change.

[1] Geospatial Data Solutions Center, University of California, Irvine, CA 92697, USA. [2] Faculty of Natural Sciences, Universidad del Rosario, Bogotá, Colombia. [3] School of Earth and Environmental Sciences, Cardiff University, Cardiff, Wales, UK. [4] Biospheric Sciences Laboratory, NASA Goddard Space Flight Center, Greenbelt, MD 20771, USA. [5] Department of Earth System Science, University of California, Irvine, CA 92697, USA. ✉email: stijn.hantson@urosario.edu.co

Climate and socio-economical drivers have modified global fire regimes considerably over the past several decades and are expected to intensify in the near future[1,2]. Ensuing changes in wildfire activity, including increases in megafires within temperate ecoregions, have far-reaching consequences for vegetation dynamics, biodiversity, and carbon stocks, influencing livelihood of people by means of fire effects on multiple ecosystem services[3]. California, like many other areas worldwide, has experienced an increase in devastating fires over the past several decades[4–7]. While wildland fire is a natural part of many ecosystems in California[8], recent fire events have far surpassed historical norms in terms of fire extent, intensity and severity. Mean annual burned area by Californian fires has more than doubled over the last decade, increasing from $1721\,km^2\,yr^{-1}$ during 1980–2010 to $3309\,km^2\,yr^{-1}$ during 2010–2018[9]. Considerable research has been conducted regarding the drivers of this recent increase in burned area. Most of this research has focused on the impact of historical fire suppression and climate change as possible causes of this recent increase[4,5,10–12]. During the early and middle part of the 20th century firefighting contributed to a reduction of area burned, which has led to a buildup in fuels in many areas[13]. The increase in fuel density may be driving some of the observed increase in burned area[11,12]. However, other analysis indicates that the increases in fuels alone cannot explain the observed increase in burned area. Temperatures have increased across California in recent decades[14], contributing to an increase in vapor pressure deficit[4]. As vapor pressure deficit (VPD) is a key factor determining dead fine fuel moisture content and correlates well with annual burned area in many drought prone ecosystems, climate change is also likely contributing to the increase in burned area in forested ecosystems[4]. A warming climate has also lengthened the fire season in spring and fall[10]. Further, increases in population and areas in the wildland urban interface (WUI) has increased fire occurrence and expanded the climate niche under which fires can occur[15]. Interpretation of the impacts these rapid changes in wildfire drivers also requires consideration of processes evolving on much longer timescales, over a period of centuries to millennia. Fires occurred frequently in California before Euro-American settlers arrived, ignited by lightning and indigenous people for ecosystem management[8]. Although burned area over the past decade was higher than during any other decade in the historical record maintained by the California Department of Forestry and Fire Protection (Cal-FIRE), present day burned area is still likely below levels that occurred during middle and late stages of the Holocene for many Californian ecosystems[16]. However, historical fire incidence was characterized by low severity fires, with high severity fires being rare across most of California ecosystems[17]. This contrasts with the contemporary fire regime, in which high intensity and high severity fires cause firefighter and civilian fatalities and contribute to damages to both infrastructure and ecosystems[18].

Many studies have tried to disentangle the drivers of the recent increase in burned area across California; less work has examined the factors changing fire behavior patterns and impact. Overall, the impact a fire has on the environment depends on how living organisms cope with the energy released by combustion of fuel[19]. Fire intensity can be estimated by fuel type, fuel amount consumed, and the spread rate of the fire[20]. As such, fire behavior determines the impact fire has on the environment[21], and studies solely focusing on burned area give an incomplete picture of the recent changes in fire activity. The current focus on burned area is largely driven by the high-quality burned area datasets, which are readily available, while only limited information regarding other fire characteristics exist[22]. Furthermore, little quantitative information is available over large areas on how fire behavior impacts fire severity and tree mortality.

The improved 375 m spatial resolution and 12-h global coverage of the active fire detections from the VIIRS instrument onboard the Suomi National Polar-orbiting Partnership (Suomi-NPP) satellite allow for precise mapping of large wildfire perimeter locations starting in 2012[23]. Here, we use these observations to develop a novel dataset of daily fire spread for all large, multiday fires in California during 2012–2018 (see methods; Supplementary Fig. 1). The dataset includes daily rate-of-spread for 214 individual fires including 2939 fire days (sum of the days with active fire detections for each fire) and covering $21,558\,km^2$ of burned area. As a function of ecoregion, the dataset covers $13,699\,km^2$ in Northern California and $7,859\,km^2$ in Mediterranean California. Fire sizes represented in the dataset range from 4 to $1660\,km^2$ (Supplementary Fig. 2). The dataset captures a wide range of daily rate-of-spread values, which vary from 0 to $22.6\,km\,day^{-1}$. The fires represented here are only fires that were not contained after the first day of fire spread, and hence only include a small fraction of the total number of reported fires (8.3%). Nevertheless, this set of large fires represent 88% of the total area burned in California during this period[9]. The ignition source for each fire was extracted from the California Fire Resource Assessment Program (FRAP) fire perimeter database[9]. Lightning ignitions are the cause of 42.1% of the fires and 44.4% of the burned area in our rate-of-spread database, while human-caused fires represent 39.3% of fires and 33.7% of burned area (Supplementary Fig. 3). We note there is a residual set of fires that contribute to total fire number (18.6%) and burned area (21.9%) for which the cause is unknown; this set of fires was excluded from any analysis for which we compared the impact of the ignition source.

We use this fire rate-of-spread dataset here to address a grand challenge in fire science: to link meteorology and ecosystem status at the time of ignition with fire behavior, and concurrently, fire behavior with fire severity and post-fire ecosystem impacts. While in past work there has been exploration of the processes structuring relationships among individual linkages, much less work has been done to explore dynamics along the full length of the causal chain. Moreover, it is unclear how human and lightning-caused fires lead to different outcomes as a consequence of diverging interactions through this chain. It is often assumed that large wildfires caused by human activity are more devastating than lightning-caused fires, and here we hypothesize that this may occur because of an increased probability of human ignition during periods of extreme fire weather. To address this hypothesis, we ask the following four questions. First, do human-caused fires occur under different environmental conditions than lightning-caused fires? Second, do such differences in environmental conditions contribute to differences in fire behavior between human and lightning-caused fires? Third, does fire rate-of-spread influence fire severity? Fourth, do human-caused fires have a different impact on Californian ecosystems than lightning-caused fires? Our results support the hypothesis, indicating that human ignitions indeed tend to coincide more with extreme weather conditions. As a result, human-caused fires spread faster during the first few days after ignition. These faster, more intense fires, in turn, contribute to more severe ecosystem impacts.

## Results

**Fire expansion**. We observe significant differences in daily fire size between human-caused and lightning-caused fires during the first several days after ignition. Human fires are on average 6.5 times larger at the end of the first day compared to lighting caused fires (18.8 vs. $2.9\,km^2$; Welch Two-Sample $t$-test, $t = 6.51$, df = 169, $p = 8.27e^{-10}$; Fig. 1 and Supplementary Fig. 4). On subsequent days, human-caused fires increase in size faster than

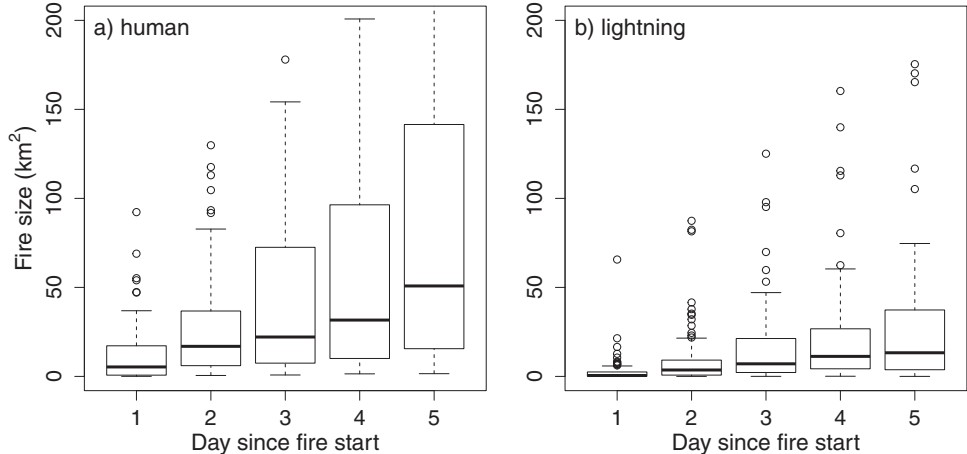

**Fig. 1 Initial fire size during the first 5 days after ignition.** Boxplots of the fire size at the end of day 1 through day 5 after ignition for fires caused by humans (**a**) or lightning (**b**) in California. The y-axis is cutoff at 200 km$^2$ to enable visualization of median differences; the full figure is shown as Supplementary Fig. 4. Differences between the two fire types are significant for each day (Welch Two-Sample t-test, day 1: $t = 6.51$, $p = 8.27e^{-10}$; day 2: $t = 6.98$, $p = 1.20e^{-10}$; day 3: $t = 5.37$, $p = 3.29e^{-7}$; day 4: $t = 4.62$, $p = 9.47e^{-6}$; day 5: $t = 4.93$, $p = 3.089e^{-06}$). The sample size $n = 82, 79, 67, 58, 44$ from day 1-5 in panel **a** and $n = 90, 82, 77, 67, 64$ from day 1-5 in panel **b**. The Boxplot represent the first quartile, the median, and the third quartile as a box, with the whiskers denote the minimum and maximum if within the range of the first quartile $-1.5\times$ the interquartile range and the third quartile $+1.5\times$ the interquartile range, while outliers are represented as points.

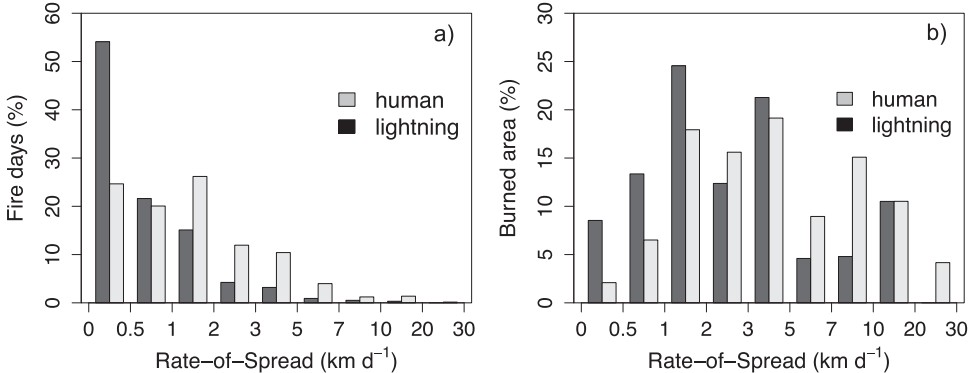

**Fig. 2 Frequency distribution of daily fire rate-of-spread.** Percentage of fire days with a given fire spread rate (**a**) and percentage of burned area with a given fire spread rate (**b**), as a function of fire type. A fire day is a day on which active fire detections were present for a given fire. Fire spread rate is defined as the 95th percentile of the spread rate over all active fire detections on the fire line. The mean daily rate-of-spread is 1.83 km day$^{-1}$ for the human-caused fires and 0.84 km day$^{-1}$ for lightning-caused fires shown in panel **a**. Similarly, mean rate-of-spread normalized by burned area (shown in panel **b**) is 5.6 km day$^{-1}$ for the human-caused fires and 3.5 km day$^{-1}$ for lightning-caused fires. The frequency distributions of daily rate-of-spread between human and lightning-caused fires as a function of fire day or burned area are significantly different.

lightning-caused fires, with human-caused fires reaching on average a size that is more than 3-fold larger than lightning-caused fires after 5 days (115.1 vs. 37.5 km$^2$, Welch Two-Sample t-test, $t = 4.93$, df $= 104$, $p = 3.^{09e-6}$). As human-caused fires are more frequent in Mediterranean California (Supplementary Fig. 2) this could reflect geographic differences in climate and fire behavior. However, the observed differences in fire size on day 1 between human-caused and lightning-caused fires is robust when we separately assess the relationship for either Northern California or Mediterranean California ecoregions, or for summer and fall periods (Supplementary Fig. 5). Even when considering only high forest biomass areas (>150 Mg ha$^{-1}$) within the Northern ecoregion we observe a significant difference in fire size between human and lightning-caused fires for the first 5 days following ignition (Supplementary Fig. 6).

The differences in initial fire sizes originate from differences in fire spread rates between human and lightning-caused fires. We estimate daily fire rate-of-spread as the 95th percentile of the distance between each active fire detection along the active fire line and the previous day's fire line (Supplementary Fig. 1). Daily rate-of-spread follows a skewed distribution, characterized by a high frequency of fire days with low spread rates (<0.5 km day$^{-1}$) and infrequent fire days with fast spread rates (>5 km day$^{-1}$; Fig. 2 and Supplementary Fig. 7). The distributions of rate-of-spread are significantly different between human and lightning-caused fires (Two-sample Kolmogorov–Smirnov test, $D = 0.35036$, $p = 2.2e^{-16}$), where, on average, human-caused fires are faster (1.83 km day$^{-1}$) than lightning-caused fires (0.84 km day$^{-1}$). Days where fires move slowly contribute relatively little to the overall burned area, so that the mean fire spread rate normalized by burned area is higher (see methods), with a mean of 5.6 km day$^{-1}$ for human-caused fires and 3.5 km day$^{-1}$ for lightning-caused fires (Fig. 2). As a function of burned area, the frequency distributions of the two fire types are also significantly different (two-sample Kolmogorov–Smirnov test, $D = 0.19921$, $p = 2.2e^{-16}$). Faster spreading human-caused fires would mean that, in the absence of suppression or fire management, human-caused fires have the potential to become orders of magnitude larger than

lightning-caused fires over the full duration of a fire's lifetime. However, this is not what we observe, with the final size of human-caused fires only being somewhat larger than lightning-caused fires (113 km² vs. 84 km²). This is because human-caused fires tend to spread for a shorter period of time (Supplementary Fig. 8), with 50% of the human-caused fires taking 3 days or less to reach 75% of the final fire size, compared to 10 days for lightning-caused fires.

**Drivers of fire growth**. The observed differences between human- and lightning-caused fires are shaped by different environmental conditions at the time of ignition and during initial phases of fire expansion. Human-caused fires occur under significantly more extreme fire weather than lightning-caused fires. For example, human-caused fires are ignited during periods with higher potential evapotranspiration and on windier days (Table 1). These conditions are likely to reduce dead fuel moisture and accelerate fire spread rates[24], allowing human-caused fires to grow faster than lightning-caused fires. Furthermore, human-caused fires are ignited in areas with lower tree cover and hence lower live biomass, with fires spreading faster through dry grass and shrub fuels than denser and moister forest fuels. These findings are robust when taking ecosystem and seasonal patterns into account (Supplementary Tables 1 and 2).

**Table 1 Mean environmental conditions on the ignition day for human- and lightning-caused fires in FRAP across California (2012–2018).**

|  | Human | Lightning | p |
|---|---|---|---|
| Energy release component (-) | **67.6** | 64.4 | 1.3e⁻⁴ |
| Potential evapotranspiration (mm day⁻¹) | **8.6** | 7.6 | 3.7e⁻¹⁶ |
| Windspeed (m/s) | **3.3** | 2.6 | <2.2e⁻¹⁶ |
| Forest biomass (Mg/ha) | 39 | 131 | <2.2e⁻¹⁶ |
| Forest cover (%) | 25 | 63 | <2.2e⁻¹⁶ |

Significant differences for human- and lightning-caused fires were assessed using the Welch Two-Sample *t*-test. Data for all variables assessed and for spatial and temporal subsets are presented in Supplementary Tables 1 & 2. The comparisons here are for the complete FRAP dataset, representing a larger set of fires than in the fire rate-of-spread dataset.
Bold values indicate the values with the highest fire risk for each meteorological variable with a significant difference between the two fire types (*p* < 0.01).

**Fire impact**. Human modification of fire occurrence in time and space has important consequences for how wildfire affects ecosystem function. Fire fronts that move faster release energy at a faster rate[20]. Greater energy release is likely to raise temperatures near the surface to higher levels, inflicting more damage to the cambium of trees with thin or damaged bark[25]. Hotter fires are also more likely to generate fires with higher flame heights that jump into the overstory, becoming a crown fire and killing trees by means of damages to the canopy overstory[25]. As a consequence, faster moving fires may have a larger impact on ecosystems[22]. Here we indeed observe a statistically significant relationship between fire rate-of-spread and tree mortality (Fig. 3). Slow moving fires (mean daily fire spread <0.5 km day⁻¹) result in low tree mortality rates (mean 15.3 ± 18.0%). In contrast, fast spreading fires (>2 km day⁻¹) result in a threefold increase in tree mortality rates (mean 48.3 ± 19.7%). This relationship is similar for both human and lightning-caused fires (Fig. 3). The relation between fire rate-of-spread and fire severity is not only robust for tree mortality but is observed for other remotely sensed fire severity indicators, including difference normalized burn ratio (Supplementary Fig. 9). As human-caused fires show on average a higher rate-of-spread, we observe that human-caused fires contribute to significantly higher tree mortality than lightning-caused fires (Table 2).

**Extreme fires**. Extreme fires influence fire statistics in a disproportional way. The top 10% of days with the fastest fires result in 55% of burned area (Fig. 2). Since the fastest moving fires induce higher levels of tree mortality (Fig. 3), a relatively small reduction of fire activity during extreme weather conditions could lead to a significant reduction in fire-induced tree mortality across California.

## Discussion

Humans have modified the fire regime across California in a myriad of ways. One of the most notable, besides fire exclusion, is the increase in ignition frequency, either directly by human agents or indirectly from damages to infrastructure such as a downed powerline during a windstorm. Previous work has shown that this increase in ignition incidence has widened the environmental niche under which fires can occur[15,26], and this finding is consistent with the monthly distributions of large fires for California shown in Supplementary Fig. 10. Our fire rate-of-

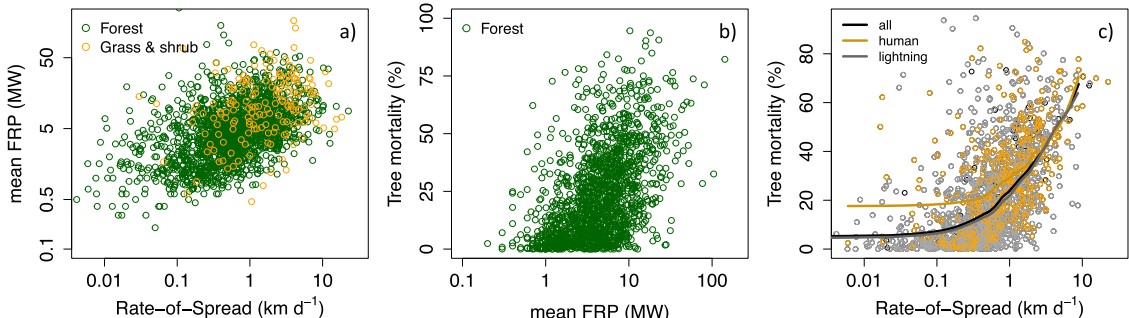

**Fig. 3 The cascade from fire behavior to fire intensity and tree mortality. a** Relationship between daily fire radiative power (FRP; mean along fire line) as a function of daily fire rate-of-spread (*F*-test, *F*-statistic = 504.6, *N* = 1722, *R*² = 0.23, *p* < 2.2e⁻¹⁶). **b** Relationship between mean FRP and tree mortality (% reduction in basal area) (*F*-test, *F*-statistic = 511.3, *N* = 1556, *R*² = 0.25, *p* < 2.2e⁻¹⁶). **c** Relationship between tree mortality (% reduction in basal area) and daily fire rate-of-spread (*F*-test, *F*-statistic = 563.5, *N* = 1556, *R*² = 0.27, *p* < 2.2e⁻¹⁶). The running means in panel **c** are plotted for all fires (black line) and separately for human-caused fires (orange line) and lightning-caused fires (gray line). Plots for other fire severity indicators are presented in Supplementary Fig. 9. Panels **b** and **c** include data across Californian forests for each fire day where the fire expanded more than 1 ha. Linear regression was used to estimate *R*² and *p*-values. Fire rate-of-spread is calculated as the 95th percentile spread rates along the full active fire perimeter fire on that day. The apparent difference at low rate-of-spreads between human-caused and lightning-caused fires in panel **c** is likely statistically insignificant given the low density and high scatter of available data under these conditions.

**Table 2 Differences in fire severity between human-caused and lightning-caused fires.**

| | Ecoregion | dNBR | | rdNBR | | Tree mortality (%) | |
|---|---|---|---|---|---|---|---|
| | | Human | Lightning | Human | Lightning | Human | Lightning |
| Maximum | Northern | 738* | 665* | 750** | 666** | 92.7* | 85.6* |
| | Mediterranean | 541** | 444** | 536* | 461* | 94ᵃ | 95ᵃ |
| Mean | Northern | 332** | 271** | 344** | 272** | 44.4* | 35.2* |
| | Mediterranean | 270 | 252 | 265 | 269 | 63* | 78* |

Maximum impact is estimated as the 95th percentile of each fire severity metric considering the distribution of each metric across the area burned for each fire type. Statistics are presented separately for Northern and Mediterranean California ecoregions (Fig. S3). Severity was assessed using the difference normalized burn ratio (dNBR), the relative difference normalized burn ratio (rdNBR), and tree mortality (percentage reduction in tree basal area). Significance is measured by the Welch Two-Sample *t*-test.
*$p < 0.05$, **$p < 0.01$.
ᵃToo few data to assess significance.

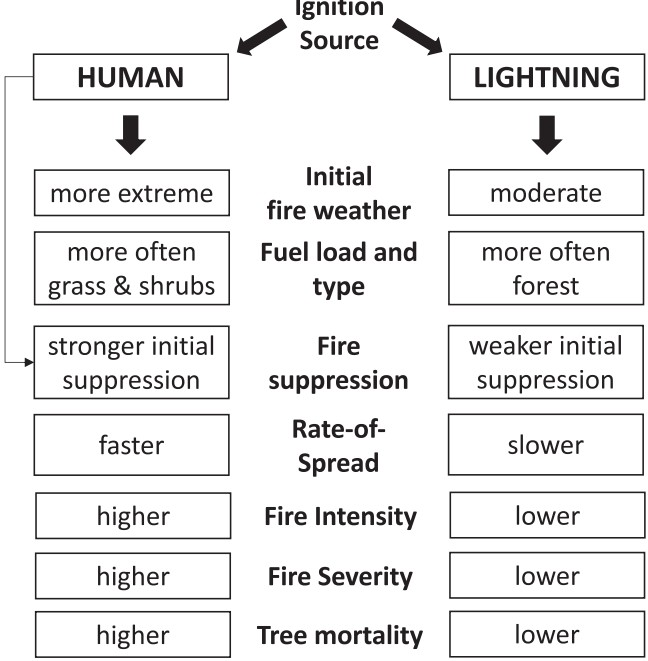

**Fig. 4 Conceptual scheme linking ignition type to fire behavior and post-fire impact.** Ignition source influences the probability of ignitions coinciding with extreme fire weather, and how this propagates through to fire behavior, intensity, and resulting ecosystem impacts. Human-caused ignitions occur throughout the year but are often extinguished quickly due to their proximity to people and infrastructure. Hence, human-caused fires can normally only grow large under more extreme fire weather conditions, which allows for high initial rate-of-spread that increases the likelihood fires can escape initial suppression. In contrast, lightning-caused fires generally coincide with more moderate fire weather but are often far from any infrastructure or human presence, which limits the effectiveness of initial suppression. As a consequence, human-caused fires tend to be faster and of higher intensity, which leads to higher fire severity and tree mortality when compared to lightning-caused fires.

spread dataset allowed us to explore other aspects of the fire regime humans have altered, including interactions between meteorology, wildfire behavior, and ecosystem impacts. We find that the large number of human ignitions throughout the year significantly increases the likelihood of ignitions during periods of extreme weather (Table 1 and Fig. 4)[27]. Human-caused fires under less extreme fire weather conditions spread slowly and can often be rapidly extinguished, and as such, barely contribute to the overall burned area. However, fires that start during extreme fire weather conditions are more challenging to contain[28]. These

rapid starts can result in very large active fire lines, driving further expansion on subsequent days (Fig. 1). While lightning-ignition can occur under similar extreme fire weather conditions, lightning storms usually are not associated with such weather patterns; instead, these coincide frequently with higher levels of atmospheric humidity and localized rainfall events that limit fire spread (Table 1 and Supplementary Tables 1 and 2). As such, lightning-caused fires generally expand at a slow speed during their initial growth phase (Figs. 1 and 2). Containment of these fires is limited partially by accessibility, as these ignitions are often far from any infrastructure. As a result, these fires can potentially grow into larger fires, with limits posed by natural discontinuities in fuel structured by topography. More importantly, the current "let burn" policy allows lightning-ignited fires to burn under certain conditions[29]. On the contrary, most human-caused fires are vigorously suppressed and anthropogenic landscape fragmentation further limits fire, resulting in shorter fire duration for human-caused fires than for lightning-caused fires (Supplementary Fig. 8). An important next step should be to determine the degree to which fire suppression resource allocation differs for human and lightning-caused fires and how this propagates to fire growth and fire damage.

The fire weather on the day of ignition is an important driver of rate-of-spread (Table 1 and Supplementary Tables 1 and 2), with fire events occurring during extremes in fire weather more likely to escape early containment. This finding is in line with previous literature on meteorological drivers of fire spread[30], but limited field observations exist that show how these relationships structure day-to-day variations in spread rate. Surprisingly, we found that spread rates are strongly skewed, with most days showing slow spread rates and rare instances of very fast spread rates (Fig. 2). A similar distribution in spread rates has been observed over the boreal region[31]. This is in contrast to the representation of fire in some models, which tend to overestimate spread rate under average fire weather conditions likely from an incomplete representation of the non-linear and additive relationships between fire weather and fire spread[31]. The high variability in fire spread rate likely originates from complex non-linear interactions between fire weather, fuels, and topography, which are often not fully captured in modeling approaches[25].

Fuel type was also an important driver of initial fire spread. Human-caused fires occurred more frequently in ecosystems with lower forest cover and aboveground biomass, compared to the much higher biomass densities in the mid-elevation forests where most lightning ignitions occur. As fine fuels from grass and shrubs rapidly dry out during dry summer conditions and have a high surface area to volume ratio, these fuels can support very fast-moving fires. Forests, in contrast, often contain a large amount of live fuels that have a higher moisture content, thus limiting rates of fire spread. As fuel type at the start of ignition

seems to be an important factor determining initial fire spread, fuel treatment around areas with high ignition probability may offer an opportunity for mitigation of future damage and risk of extreme fire events.

Management options critically depend on the local ecosystem characteristics and social context. While our findings are robust when separately considering either northern or Mediterranean California ecosystems, these ecosystems have important structural and functional differences. For example, the importance of shrub vegetation in driving fire behavior in Mediterranean California leads to faster fire spread rates than in northern ecosystems. Furthermore, Mediterranean California has a low frequency of lightning-caused fires and is more densely populated. Hence, management strategies will necessarily need to take such differences into account.

Fires can exert a broad array of ecosystem effects, with the magnitude of the impact depending on fire behavior and intensity. Tree mortality, for example, is often estimated in models as a function of crown scorch fraction and stem char height, which, in turn, are based on fire line intensity, the residence time of fire exposure, and flame height[25]. However, the drivers of fire-induced tree mortality are not well understood. One problem is that neither fire behavior nor ecosystem impact can be easily measured in the field at large spatial and temporal scales. New remote sensing-based wildfire tracking capability from VIIRS[23] allowed us to empirically quantify how fire behavior is related to fire severity, and to identify that fire severity and tree mortality is higher for faster moving fires. Increases in FRP observed for the fastest moving fires suggest that the underlying mechanism is tied to greater energy release rates in fast-moving fires that increase plant exposure to higher surface temperatures. While it has been shown previously that fire weather and fuels drive fire severity[32], our results show that these environmental drivers impact fire severity through their impact on fire spread rate and fire intensity. Indeed, fire rate-of-spread is required as an input to calculate one of the most widely used fire intensity indices (Byram's fireline intensity[20]). Hence our observations may provide the basis for improving our understanding of the physical drivers of fire severity and fire-induced tree mortality and the representation of fire processes in ecosystem models[33,34].

Our results suggest that the current "let burn" policy of natural occurring lightning-caused fires might result in lower burn severity and tree mortality compared to human-caused fires and hence may be an effective way to restore fire as an essential aspect of many Californian ecosystems. On the contrary, we observe an emerging human-ignited fire syndrome, where ignitions coincide with extreme fire weather. These fires spread fast and cause a significant enhancement of ecosystem impacts and tree mortality compared to levels expected from a lightning-driven fire regime. This anthropogenic change in fire regime has consequences for programs intended to increase carbon storage in natural ecosystems, air quality, and ecosystem services like freshwater supply. As days with extreme fire weather conditions will become more frequent[35], reducing ignitions during most extreme weather conditions will therefore be a key mitigation strategy under scenarios of future change.

## Methods

**Fire rate-of-spread dataset.** We estimated fire rate-of-spread (km day$^{-1}$) for California fires represented in the FRAP database[9] between 2012 and 2018 with a size larger than 300 ha and a duration of 2 or more days. The FRAP database is maintained by the State of California and updated annually. The effort, led by CalFIRE, integrates information on fire perimeters from the US Forest Service, the Bureau of Land Management, and the National Park Service. Additional fire perimeters are entrained from other US federal agencies when available (e.g., the Dept. of Defense and Bureau of Indian Affairs)[9]. The fire perimeters are digitized from multiple information sources including ground or aircraft global positioning

system (GPS) coordinates, infrared imagery, photo interpretation, and mixed data sources. Key attributes associated with each fire include the start date, containment date, and the cause of ignition. Here we used the FRAP dataset to identify the outer final perimeter of all large wildfires greater than 300 ha and to identify the cause of ignition. Our thresholds for building a multiday fire spread dataset (greater than 300 ha and 2 or more days of burning) were set by accuracy requirements from our use of the alpha-hull algorithm with the VIIRS active fire observations (described below) and far exceeded the minimum fire size requirements for the FRAP database. We used the ignition source for each fire in FRAP to separate between human, lightning, and unknown ignition causes. Data from fires with unknown ignition source were excluded from all analysis where we separated the results based on ignition source.

For each fire, we extracted the 375 m I-band active fire observations from the Visible Infrared Imaging Radiometer Suite (VIIRS) sensor onboard the Suomi-National Polar-orbiting Partnership (S-NPP) satellite[23]. All active fires within a 750 m buffer around the FRAP perimeter were extracted within the period between fire ignition and containment from the FRAP database, with a 1-day temporal buffer. If the ignition or containment date was missing from the FRAP polygon, a 3-month time window for active fire extraction was applied. Fire growth was estimated each day by grouping all active fire detections occurring from the daytime overpass (approximately 1:30 p.m.) and following nighttime overpasses (approximately 1.30 a.m. on the next day). In the case of overlapping swaths, we used the last overpass as the time of fire detection. Based on these active fire data we perform a best estimate of the fire perimeter at the end of the day, using the center of each active fire pixel as best guess of where the fire was located. The fire perimeter is delimitated using the alpha-hull approach, a generalization of the convex hull[36], where a convex hull is the simplest shape that embeds all data points. The alpha-hull extends upon the convex hull, allowing for more complex shapes which embed all points, where the complexity is determined by the parameter alpha[37]. In our case, we attempted to capture as accurately as possible the complex shapes of fire lines without breaking the fire up into multiple parts. To accomplish this, we chose an alpha value of 0.05 units, as lower values led to discontinuous perimeters. In cases where a single continuous perimeter could not be obtained with this parameter setting, the alpha value was increased by 0.05 units. We use the R package "alphahull"[38] to generate the polygon around the active fire detections. We consecutively estimated a new fire perimeter for each timestep for which active fire data were available. Fires caused by lightning often have multiple ignitions. Groups of active fire data were considered separate fires when the minimum distance between a point and any other points within the active fire groups was larger than 1500 m, and individual polygons were drawn for each group of active fire detections. When a fire merges the two groups of active fires detections are considered as one and a polygon is drawn around all of them together from that timestep onwards. A schematic outlining how the fire spread data was constructed is provided in Supplementary Fig. 1. A comparison with daily fire growth for 14 fires is presented in Supplementary Methods 1; Supplementary Fig. 11 & Supplementary Table 3).

For each day, we extracted the active fire data points that form the active fire line, including those that fall slightly behind the estimated perimeter (100 m). For each point, we estimated the minimum distance to the fire perimeter of the previous timestep and the time period between the time steps to calculate rate-of-spread (Supplementary Fig. 1). Rate-of-spread could not be estimated for the first day of fire occurrence as the exact time and location of the ignition point was normally not known. Rate-of-spread values of zero occurred when an active fire was detected, but the active fire line did not advance between timesteps.

**Analysis.** We used the daily fire rate-of-spread database generated here from 2012–2018 to assess differences in fire spread rate between human-caused and lightning-caused fires. Differences in the mean and distribution of the fire spread rate (Fig. 2) were assessed both with and without normalization by burned area. To normalize by burned area, we applied the following equation:

$$\frac{\sum_i^n BA_i * RoS_i}{\sum_i^n BA_i} \qquad (1)$$

where $BA_i$ is the burned area on day i, $RoS_i$ is the fire spread rate at day $i$, and $n$ is the total fire days considered.

The differences in mean fire number, size and rate-of-spread between human and lightning-caused fires were assessed using the Welch Two-Sample $t$-test. As many of the fire behavior or size variables had heavy-tailed distributions, these were log-transformed before assessing the significance of differences between the two different fire types. We tested whether the distributions in daily fire rate-of-spread are significantly different between human and lightning-caused fires using the Kolmogorov–Smirnov test. Variance in rate-of-spread is given by the interquartile range (IQR).

We analyzed the difference in climate and environmental factors at the day of ignition between human-caused and lightning-caused fires based on all fires represented in the FRAP database, using the fire cause as indicated in the FRAP database, considering that all fire causes related to human activity are human-ignited fires. Variables used to assess the environmental and fire weather conditions at time of ignition between human and lightning-caused fires were

the vegetation biomass dataset from Oregon State University Landscape Ecology, Modeling, Mapping & Analysis (LEMMA) for the year 2010 at 30 m spatial resolution[39] and the daily climate data as provided by gridMET at 4 km spatial resolution[40].

To assess the relationship between rate-of-spread and fire severity, we used remotely sensed fire severity indexes, including differenced normalized burn ratio (dNBR) and relative differenced normalized burn ratio (rdNBR) as generated by the Monitoring Trends in Burn Severity (MTBS) project using Landsat data[41]. To have a fire severity indicator more directly related to ecosystem structure, we also use the tree mortality product, quantified as the percent reduction in tree basal area after fire, generated by the Forest Service[42], available at: https://www.fs.usda.gov/detail/r5/landmanagement/gis/?cid=STELPRDB5327833 (last accessed 13/02/2020). For every fire, we sampled the fire severity for the area the fire expanded into each day. Fires or subsets of fires not represented in the MTBS or tree mortality datasets were excluded from the analysis. For each day, we calculated the mean and 95th percentile of fire severity indicators based on the 30 m pixels. We also summarized fire severity indicators for each fire to assess whether there are differences between human and lightning-caused fires. We summarized rate-of-spread by taking the 95th percentile of the spread rate over all active fire detections on the fire line as a conservative estimate of maximum rate-of-spread for that day. We also calculated the mean rate-of-spread across the entire fire front to show the robustness of the results and found that the mean and maximum fire spread rates are highly correlated (Supplementary Fig. 12).

As California ecosystems are highly diverse, we performed our analysis separately for two large ecosystem groups with extensive fire occurrence in California, namely the level 2 ecoregion Mediterranean California and Northern California, which incorporates the level 2 ecoregions Western Cordillera and Marine west coast forest (Supplementary Fig. 2). To analyze the fire ecosystem impact and tree mortality, we separated forest and non-forest areas using the CALVEG vegetation map for the year 2011[43].

**Reporting summary**. Further information on research design is available in the Nature Research Reporting Summary linked to this article.

## Data availability

The California fire growth dataset generated in this study has been deposited under accession code https://doi.org/10.5281/zenodo.4248662 and will be updated regularly. The FRAP fire perimeters are available at https://frap.fire.ca.gov/frap-projects/fire-perimeters/. The GridMET climate dataset is available from http://www.climatologylab.org/gridmet.html. The tree mortality product generated by the Forest Service[42] is available at: https://www.fs.usda.gov/detail/r5/landmanagement/gis/?cid=STELPRDB5327833. The vegetation biomass dataset from Oregon State University Landscape Ecology, Modeling, Mapping & Analysis (LEMMA) is available at: https://lemmadownload.forestry.oregonstate.edu. The MTBS fire severity data is available at: https://www.mtbs.gov.

## Code availability

The statistical analyses were all performed in R 4.0.3. The code to generate the fire growth dataset can be found here: https://doi.org/10.5281/zenodo.6362832

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

## Acknowledgements
We received funding support from a University of California National Laboratory Fees grant to J.T.R. and M.L.G., a California Strategic Growth Council award to M.L.G., and funding from UC Irvine to the Center for Geospatial Data Solutions for Climate and the Environment.

## Author contributions
S.H. created the fire spread database; S.H. performed the analysis and interpreted the results with M.L.G., N.A., J.T.R. S.H. and J.T.R. wrote the manuscript with input from all coauthors.

## Competing interests
The authors declare no competing interests.
