## [Peer Review File · Nature Communications]

Human-ignited fires result in more extreme fire behavior and ecosystem impactsReviewers' Comments:

Reviewer #1:

Remarks to the Author:

Hantson et al. describe differences between human- and lightning-ignited fires in California based on new remote sensing dataset, VIIRS.

Even though their major three questions (L80-L83) are clear and import in the development of fire science area. However, I believe there are several reasons why, in its current form, it does not meet the criteria for publication in Nature Communications.

I found two critical points that should be addressed in this manuscript.

1. Separating human- and lightning-ignited fires

I was wondering the quality of ignition data and authors did not describe elsewhere in the manuscript. Authors just mentioned, "in our database (L67)". What exactly is your database to separate human- and lightning ignitions (L67)? How about the confidence level of this metadata?

2. Environment factors

Authors said "Human caused fires occur under significantly more extreme "fire weather" conditions than lightning fires", but I think fire can have higher speed due to environmental conditions (Table 1). I could not find any analysis to address solely effect of ignition type by removing the effect from environmental conditions. Indeed, the causal relationship is not clear. I doubt fire spread speed would not much different when authors compare fire cases with similar environmental conditions even though types of ignitions differ.

Reviewer #2:

Remarks to the Author:

Human-ignited fires are faster, hotter and kill more trees in California forests

Comments:

Human-ignited fires are faster and hotter, but I am not sure whether they may kill more trees if these fires occur more on grassy and shrubby landscapes instead of the dense forested area (Line 123-125).

I think this is a very well-designed research and their findings are interesting and might have potential application in fire management for the specific area (i.e. California). However, I wonder whether the comparison between human- and lightning-caused fires would be as significant if the fuel compositions are similar (e.g. dense forested area). As lightning-caused fires are hard to occur in the grassy landscape (Latham and Williams, 2001*; also shown in this study), comparing human- and lightning-caused fires prevalent in different fuel types may not reveal the differences between the two causes as much as one may have expected. Because human-caused fires usually start from the surface fuel and lightning fires ignite through the duff layer in the soil, human-caused fires would require more severe fire weather for it to occur. This should be true for fires occurred in the forested area.

Topography may limit fuel continuity, and human activities may fragment the landscape close to human residential areas. These factors could also limit fire sizes. These should be part of the discussions.

Minor comments:

(1) Line 65: total area burned;

(2) Line 76-77: evidence to support the assumption?

- (3) Line 100-102: a concept called spread day (Podur and Wotton 2011 (IJWF); Wang et al. 2014 (GCB)) may be used to explain this.
- (4) Line 104-106: I have a hard time understanding what you are trying to say here.
- (5) Line 108-113: number of spread days determines the size of fires (e.g. Wang et al. 2020 (ERL)). Other limitations could come from fuel fragmentation due to either topographic features or human activities.
- (6) Line 121-126: see comments to Line 108-113.
- (7) Line 224: fire detections--- do you mean hot spots?
- (8) Figure S9: legend should be consistent with the rest.

*<https://www.sciencedirect.com/science/article/pii/B9780123866608500131?via=ihub#aep-abstract-id7>

Reviewer #3:

Remarks to the Author:

I read the paper by Stijn Hantson and colleagues with enthusiasm, as the subject is of great interest to me. After several very active fire seasons in California, this paper is sure to be of interest to the greater scientific community. I don't see any flaws in methodology and interpretation, but I think there are some clarifications needed in the Methods and some further text in the Discussion. See specific comments for ideas to improve the paper. Most of my comments are fairly minor, but please pay special attention to comments #4, #15, and #26.

Specific comments:

1. Line 26: are fires with an unknown cause excluded from the analysis?
2. Line 215: Here you define for growth rate (km²/day) and rate-of-spread (km/day) but I don't think these terms are used consistently throughout the paper.
3. Line 235: what is a "closed perimeter"?
4. Lines 215-243 and Fig. S1: The description and illustration of how you calculated fire growth is useful. That said, fire growth is usually much more complicated, particularly with complexes with multiple ignitions. Can you please explain or show how ROS is calculated in some of the more complicated scenarios, for example, when two fires merge together? Also, how do you calculate ROS on the first day of the fire?
5. Line 251: Why is the "direction" of fire growth important?
6. Line 261: Not sure what variables are assessed and how. What does "assessed" even mean here?
7. Line 267: So, the perimeters are from FRAP and the severity is from MTBS. My guess is that MTBS data are not available for all FRAP fires and that the perimeters don't match in many/most cases. How was this reconciled?
8. Figure 1: Maybe label the panels as human and lightning so it is obvious what the panels represent without having to read the caption.
9. Figure 2. Line 410 suggests both panels should have percent of fire days on the y-axis, yet this does not seem to be the case. L
10. Figure 2, lines 415-416: is this supposed to be "fire growth", in km²/day?
11. Figure 3, line 422 (and elsewhere): These p-values are ridiculously low and you can just say something like "p<0.001". The exponent notation spit out by R is unnecessary.
12. Figure 3: Did you explore other fits besides linear? Seem like a panels b and c might be better explained by a glm with a quasibinomial distribution.
13. Figure 3, panel b: Instead of calling this "% tree mortality" here and everywhere else, you should

consider calling it what it actually is, which is "% basal area killed by fire".

14. Figure 3, panel c: It looks like the difference between human and lightning fires at low ROS is driven by only a few points. Might want to consider adding error bars or something to indicate that the differences might not be "real".

15. Figure 4 (and elsewhere like Tables 2, S1, S2, "drivers of fire growth" in the Results, and other text): Although the differences in fire weather between human and lightning fires are "statistically significant" (probably because of a large sample size), I am not convinced they are significantly different in terms fire spread and effects. Some of the differences shown in Tables 1, S1, and S2 are not really that big, so labelling them as "extreme" and "moderate" is not necessarily warranted in this figure and elsewhere.

16. Line 446: I believe you should reference Tables S1 and S2 (not Tables S2 and S3).

17. Table 1, last two rows: Pertaining to the strategy of bolding certain values, I understand how you can label fire weather as higher or lower risk, but not sure you can label biomass and forest cover as higher or lower risk. Same goes for Tables S1 and S2.

18. I'm also curious as to how much total area is burned by human and lightning fires and by ecoregion. I noted one of the figures in the Supplemental summarized by number of fires, but I'd also like to see area burned.

19. Fig. S8: "ratio", not "ration".

20. Table S2: forest cover looks like proportion and not percent.

21. Line 104-106: This sentence is confusing.

22. Lines 111-112 (and elsewhere): You can delete "in our database".

23. Line 128: "difference", not "change".

24. I may have missed it, but how are "forests" and "shrubs/grass" differentiated? Describe in Methods.

25. Line 153: Probably the most notable way in which humans have changed fire regimes is through fire exclusion.

26. Discussion: I'd like to see a paragraph or two devoted to how your findings compare and contrast to related studies that evaluated fire spread, fire severity/effects, and daily weather.

Response to reviewer comments

We thank the reviewers for their comments and suggestions and give below a detailed response in blue.

Reviewers' comments:

Reviewer #1 (Remarks to the Author):

Hantson et al. describe differences between human- and lightning-ignited fires in California based on new remote sensing dataset, VIIRS.

Even though their major three questions (L80-L83) are clear and important in the development of fire science area. However, I believe there are several reasons why, in its current form, it does not meet the criteria for publication in Nature Communications.

I found two critical points that should be addressed in this manuscript.

1. Separating human- and lightning-ignited fires. I was wondering about the quality of ignition data and authors did not describe elsewhere in the manuscript. Authors just mentioned, "in our database (L67)". What exactly is your database to separate human- and lightning ignitions (L67)? How about the confidence level of this metadata?

We thank the reviewer to point out that we did not make the source of our ignition source clear. The ignition source comes directly from the official California Fire and Resource Assessment Program (FRAP) burn area perimeter dataset. If the ignition source could not be identified, it is set as unknown. We clarify this aspect now in our manuscript and methods section. In line 66-67 we add "Ignition source for each fire was extracted from the California FRAP fire perimeter database." And we added to the methods section: "We use the ignition source for each fire in FRAP to separate between lightning, human and unknown ignition source."

The ignition source within the FRAP perimeter database comes from the official cause of ignition investigation. As each fire is a potential criminal offence, an investigator for an origin and cause determination (a qualified Wildland Fire Investigator) is assigned immediately after the detection of a fire across the US and an official investigation is undertaken to determine the ignition point and source. After the investigation determines the ignition source, this is reported to the relevant official instances. This data is recompiled by FRAP and becomes part of the FRAP fire perimeter dataset used here. This dataset is very widely used and (so far), no large mistakes have been reported. The largest uncertainty with regarding to ignition source seem to be related to the fraction of fires for which the ignition source is could not be determined.

2. Environment factors

Authors said "Human caused fires occur under significantly more extreme "fire weather" conditions than lightning fires", but I think fire can have higher speed due to environmental conditions (Table 1). I could not find any analysis to address solely effect of ignition type by removing the effect from environmental conditions. Indeed, the causal relationship is not clear. I doubt fire spread speed would not much different

when authors compare fire cases with similar environmental conditions even though types of ignitions differ.

This comment from the reviewer seems to arise in part from confusion regarding the interpretation of our results, which we do not seem to have presented clearly enough in the title, abstract, and main text of the manuscript.

We do not wish to imply that there was a causal relation between ignition source and fire behavior and impact. By definition, a fire started by a lightning or human ignition under identical environmental conditions will be identical in behavior and impact. Instead, it is the timing and location of fires which are shaped by humans, causing fires to occur under more fire-prone environmental conditions (both fuel and fire weather dimensions) and drive the differences in fire behavior and impact. Hence, it is the human behavior which contributes to a regime of human-ignited fires occurring more frequently under extreme fire weather and in locations with fuel types which are able to support high spread rates. We have now reworked key aspects of the manuscript to make this point clearer throughout the manuscript.

The main changes are:

The title was changed to “Timing and landscape position of human-ignited fires lead to more extreme fire behavior and ecosystem impacts” because of this reviewer comment.

The abstract was also modified. We added the following text to clarify how humans have impacted timing and location of fires: “We observe that human ignited fires coincide with more extreme fire weather and ignition locations in more open ecosystems. These characteristics contribute to more explosive growth in the first few days following ignition for human-caused fires as compared to lightning-caused fires.” And with regard how this propagates to influence ecosystem impact, we added: “These results provide evidence that by shaping the conditions under which fires occur, humans are considerably modifying fire dynamics and ecosystem impacts.”

Similarly, at the end of the introduction section, we rephrased and added the following research questions as follow: “First, do human caused fires occur under different environmental conditions than lightning caused fires? Second, do such differences contribute to difference in fire behavior between human and lightning caused fires?”. We did this again to clarify that we initially hypothesized that humans might have influenced fire behavior through changes in fire timing and location.

In the results, we modified the beginning of the second subsection to also clarify this. It now reads: “Drivers of fire growth. The observed differences between human and lightning caused fires are shaped by the environmental conditions at the time of ignition and during fire spread.”

Similarly, the third subsection now starts with: “Human modification of fire occurrence in time and space has important consequences for how wildfire affects ecosystem function.

Reviewer #2 (Remarks to the Author):

Human-ignited fires are faster, hotter and kill more trees in California forests

Comments:

Human-ignited fires are faster and hotter, but I am not sure whether they may kill more trees if these fires occur more on grassy and shrubby landscapes instead of the dense forested area (Line 123-125).

Shrubby and grassy areas were removed from this analysis, as mentioned in methods section: "To analyze the fire ecosystem impact and tree mortality we separated forest and non-forest areas using the CALVEG vegetation map for the year 2011". Additionally, the tree mortality is given as percentage reduction in basal area of trees, and as such our tree mortality indicator is relatively independent from tree density. We have indeed thought of calculating the absolute basal area killed but preferred not to take this step as this would mean integrating state-wide maps of basal area estimates, which are known to have large uncertainties. We have rephrased the title for clarity, and now reads: "Timing and landscape position of human-ignited fires lead to more extreme fire behavior and ecosystem impacts". We have also modified the relevant part of the caption for figure 3 to explicitly indicate how tree mortality was characterized, which now reads: "(c) Relationship between tree mortality (% reduction in basal area) and daily fire rate-of-spread"

I think this is a very well-designed research and their findings are interesting and might have potential application in fire management for the specific area (i.e. California). However, I wonder whether the comparison between human- and lightning-caused fires would be as significant if the fuel compositions are similar (e.g. dense forested area). As lightning-caused fires are hard to occur in the grassy landscape (Latham and Williams, 2001*; also shown in this study), comparing human- and lightning-caused fires prevalent in different fuel types may not reveal the differences between the two causes as much as one may have expected. Because human-caused fires usually start from the surface fuel and lightning fires ignite through the duff layer in the soil, human-caused fires would require more severe fire weather for it to occur. This should be true for fires occurred in the forested area.

We agree with the author that we have to be careful not to compare two completely different ecosystems, as that indeed could lead to wrong conclusions. That was the principal reason we already separated our results between Northern and Mediterranean California ecosystems and into summer and fall fire types. To further strengthen our results, we show below the results in fire growth for an even more homogenous subset of our data, only using data from the Northern ecoregion and which additionally burn through densely forested areas with > 150 Mg/ha vegetation biomass. We still observe significant ($p < 0.01$) differences in fire spread between human and lightning caused fires for these areas with similar vegetation characteristics (see figure below). We have now added this figure as Figure S6 and added the following sentence to the main text to further strengthen the manuscript: "Even when considering only high forest biomass areas ($> 150 \text{ Mg ha}^{-1}$) within the Northern ecoregion we observed a significant difference in fire growth ($p < 0.01$) for human and lightning caused fires (Figure S6)."

Figure S6: Mean fire size at the end of day 1 through day 5 after ignition, for fires caused by humans (solid line) or lightning (dashed line) for area having $> 150 \text{ Mg ha}^{-1}$ vegetation biomass across the Northern ecoregion.

Topography may limit fuel continuity, and human activities may fragment the landscape close to human residential areas. These factors could also limit fire sizes. These should be part of the discussions.

Agreed, we now including a sentence in the discussion indicating landscape fragmentation and how they can influence fire spread. The sentence reads as follow: “On the contrary, most human-caused fires are vigorously suppressed and anthropogenic landscape fragmentation further limits fire, resulting in shorter fire durations for human caused fires compared to lightning caused fires (Figure S8)”.

With regard to topography, we now write in the discussion on page X: “Containment of these fires is limited partially by accessibility, as these ignitions are often far from any infrastructure, so that these fires can potentially grow into larger fires as well, mainly limited by natural discontinuities in fuel like topography.”

Minor comments:

(1) Line 65: total area burned;

Corrected

(2) Line 76-77: evidence to support the assumption?

The idea that a lot of the damage by wildland fires in California is caused by humans is very widespread, and definitely the feeling one gets when reading articles about wildland fires in the press and media, although these are generally presented without empirical support. A little search on the web gives immediately titles like: “Humans are making wildfires worse...”, or phrases like “Although lightning fires cause immense damage, they account for just a fraction of the annual wildfire devastation in California... the rest is caused by human activity...”.

(3) Line 100-102: a concept called spread day (Podur and Wotton 2011 (IJWF); Wang et al. 2014 (GCB)) may be used to explain this.

Thank you for these references. We have now added an extra sentence to the discussion section and cite the Wang reference there: “A similar distribution in spread rates has been observed over the boreal region²⁶. This is in contrast to the representation of fire in most models, which overestimate spread rate

under average fire weather conditions, probably due to an incomplete representation of the nonlinear and additive relationships between fire weather and fire spread ²⁶.”

(4) Line 104-106: I have a hard time understanding what you are trying to say here.

We are sorry we didn't make ourselves clear here. The point we wanted to make is that the mean spread rate through time does not give a good indication of what happens in the field, as most burned area happens during fast spread rates. In this sentence we give the mean spread rate when normalized by burned area instead of through time. We now clarify this in the text, which now reads:

“Days where fires move slowly contribute relatively little to the overall burned area, so that the mean fire spread rate normalized by burned area is higher, with a mean of 5.6 km day⁻¹ for human caused fires and 3.5 km day⁻¹ for lightning caused fires (Figure 2).”

(5) Line 108-113: number of spread days determines the size of fires (e.g. Wang et al. 2020 (ERL)). Other limitations could come from fuel fragmentation due to either topographic features or human activities.

In our case the final fire size seems to depend also strongly on the length of the active fire line, but a formal analysis is outside the scope the current study. We now cite the reference of Wang et al (see also previous comment) in the discussion section, where we added: “A similar distribution in spread rates has been observed over the boreal region ²⁶.”. We now also include the following sentences with regard to fuel fragmentation due to topography and land use, and how this influences fire spread to the discussion section:

“On the contrary, most human-caused fires are vigorously suppressed and anthropogenic landscape fragmentation further limits fire, resulting in shorter fire durations for human caused fires compared to lightning caused fires (Figure S8).”

And:

“The high variability in fire spread rates likely originate from complex non-linear interactions between fire weather, fuels, and topography, which are often not fully captured in modelling approaches ²⁵.”

(6) Line 121-126: see comments to Line 108-113.

See comment above

(7) Line 224: fire detections--- do you mean hot spots?

Yes. There are indeed multiple names for these (e.g. fire thermal anomalies, active fires, ...). We prefer to use active fire detections as this describes best what the data is for people outside the remote sensing community.

(8) Figure S9: legend should be consistent with the rest.

We have now adapted the figure legend for consistency. We have also adapted the figure caption for clarity.

Reviewer #3 (Remarks to the Author):

I read the paper by Stijn Hantson and colleagues with enthusiasm, as the subject is of great interest to me. After several very active fire seasons in California, this paper is sure to be of interest to the greater scientific community. I don't see any flaws in methodology and interpretation, but I think there are some clarifications needed in the Methods and some further text in the Discussion. See specific comments for ideas to improve the paper. Most of my comments are fairly minor, but please pay special attention to comments #4, #15, and #26.

We thank the reviewer for their detailed and constructive comments.

Specific comments:

1. Line 26: are fires with an unknown cause excluded from the analysis?

Fires with unknown cause are indeed excluded from all analysis where we separate out between human and lightning caused fires. We now clarify this further in the methods section where we have added the following sentence: "Data from fires with unknown ignition source were excluded from all analysis where we separated the results based on ignition source."

2. Line 215: Here you define for growth rate (km²/day) and rate-of-spread (km/day) but I don't think these terms are used consistently throughout the paper.

We indeed do not use the growth rate estimates in the results presented here and have now removed this part from the methods section to avoid confusion.

3. Line 235: what is a "closed perimeter"?

In case of scattered out active fire detections, the algorithm can fragment the fire, which we avoided by setting this parameter dynamically. We now rephrased this sentence to: "In case a single continuous perimeter could not be obtained with this parameter setting, the alpha value was increased by 0.05 units."

4. Lines 215-243 and Fig. S1: The description and illustration of how you calculated fire growth is useful. That said, fire growth is usually much more complicated, particularly with complexes with multiple ignitions. Can you please explain or show how ROS is calculated in some of the more complicated scenarios, for example, when two fires merge together? Also, how do you calculate ROS on the first day of the fire?

There are indeed cases of multiple ignitions for some lightning caused fires. In those cases we do create individual polygons for each ignition until the fires merge together (when they get closer than 1.5km (see methods)). When a fire merges the two groups of active fires detections are considered as one and a polygon is drawn around all of them together. The calculation of RoS is not affected as we estimate the shortest distance for each active fire detection to the fire perimeter at the previous time step (which can be multiple perimeters in this case).

We have adapted the methods section to better explain how we treat fires with multiple ignitions, which now reads: "Groups of active fire data were considered separate fires when the minimum distance between a point and any other points within the active fire groups was larger than 1500

m, and individual polygons were drawn for each group of active fire detections. When a fire merges the two groups of active fires detections are considered as one and a polygon is drawn around all of them together from that timestep onwards.”

We indeed do not estimate ROS for the first day of fire occurrence as the exact time and location of the ignition point are normally not known. We indicated this implicitly in Figure S1, but now mention this explicitly in the figure caption of Fig S1 (“As a consequence, rate-of-spread cannot be estimated for the day of ignition.”), as well as in the methods section for clarity, where we included the following sentence: “Rate-of-spread could not be estimated for the first day of fire occurrence as the exact time and location of the ignition point are normally not known.”

5. Line 251: Why is the “direction” of fire growth important?

This is another variable we calculate within the fire growth database but is not used in this study. We now remove these sentences to avoid confusion.

6. Line 261: Not sure what variables are assessed and how. What does “assessed” even mean here?

We agree we did not explain ourselves clearly here and now rephrased this sentence as follows: “Variables used to assess the environmental and fire weather conditions at time of ignition between human and lightning caused fires were the vegetation biomass dataset from Oregon State University Landscape Ecology, Modeling, Mapping & Analysis (LEMMA) for the year 2010 at 30m spatial resolution ³² and the daily climate data as provided by gridMET at 4 km spatial resolution ³³.”

7. Line 267: So, the perimeters are from FRAP and the severity is from MTBS. My guess is that MTBS data are not available for all FRAP fires and that the perimeters don’t match in many/most cases. How was this reconciled?

We use the final daily fire growth data and intersect this with MTBS (so not directly FRAP, but indeed constrained by FRAP with and additional buffer). While for these larger fires there is a pretty good agreement between MTBS and FRAP, it is indeed not perfect. We extracted the data from MTBS (rdNBR & dNBR) for each daily fire expansion in our dataset ignoring fires or areas of fires which are not included by MTBS. We now clarify this in the methods section as follows: “Fires or subsets of fires not represented in the MTBS or tree mortality datasets were excluded from analysis.”

8. Figure 1: Maybe label the panels as human and lightning so it is obvious what the panels represent without having to read the caption.

We now include human and lightning after the a) and b). (and similarly, for figure S4).

9. Figure 2. Line 410 suggests both panels should have percent of fire days on the y-axis, yet this does not seem to be the case.

This was indeed not well explained in the caption. We have now improved the caption which now reads: “Percentage of fire days with a given fire spread rate (a) and percentage of burned area with a given fire spread rate (b), separated by ignition source.”

10. Figure 2, lines 415-416: is this supposed to be “fire growth”, in km²/day?

This confusion probably comes from not explaining ourselves at the start of the caption and we hope this is clearer now. The figure panel b indicates how much burned area occurs under a given spread rate, from which we can calculate a mean spread rate when weighted by burned area. This sentence now reads: "Percentage of fire days with a given fire spread rate (a) and percentage of burned area with a given fire spread rate (b), separated by ignition source."

11. Figure 3, line 422 (and elsewhere): These p-values are ridiculously low and you can just say something like " $p < 0.001$ ". The exponent notation spit out by R is unnecessary.

Agreed, these are all very low p values and they have been changed in the revised text to < 0.001 .

12. Figure 3: Did you explore other fits besides linear? Seem like a panels b and c might be better explained by a glm with a quasibinomial distribution.

For simplicity, we only performed linear fits, with the main objective to show that these relationships are significant. We did not have the objective to determine the best possible fit.

13. Figure 3, panel b: Instead of calling this "% tree mortality" here and everywhere else, you should consider calling it what it actually is, which is "% basal area killed by fire".

We indeed went forward and backwards with the naming this variable. We finally decided to use % tree mortality, as it is easier to understand for a wider public. We consider that the description in the caption, where we now indicate how tree mortality was defined, makes this clear for readers who want some more detail. We have now adapted the sentence in the caption regarding panel c, which now reads: "c) The relation between tree mortality (% reduction in basal area) and daily fire Rate-of-Spread ($R^2 = 0.24$, $p < 0.001$)." to avoid any confusion.

14. Figure 3, panel c: It looks like the difference between human and lightning fires at low ROS is driven by only a few points. Might want to consider adding error bars or something to indicate that the differences might not be "real".

This difference is indeed due to just a couple of data points. We tried including and uncertainty range, but the figure gets very messy, and we finally preferred to show the original data points over an indicator of uncertainty. To acknowledge this we now mention in the figure caption: "The apparent difference at low rate-of-spreads between human and lightning fires in panel c is likely statistically insignificant given the low density and high scatter of available data under these conditions."

15. Figure 4 (and elsewhere like Tables 2, S1, S2, "drivers of fire growth" in the Results, and other text): Although the differences in fire weather between human and lightning fires are "statistically significant" (probably because of a large sample size), I am not convinced they are significantly different in terms fire spread and effects. Some of the differences shown in Tables 1, S1, and S2 are not really that big, so labelling them as "extreme" and "moderate" is not necessarily warranted in this figure and elsewhere.

The relation between environmental drivers and fire rate-of-spread contain step-changes and tend to be non-linear, so that relatively modest changes can still result in relatively large changes in fire behavior. In figure 4 we chose to use moderate and extreme for two different reasons. First, under poor fire weather conditions fires will not or barely spread, so that "moderate" seem like a reasonable term to describe average fire weather/environmental conditions under which fires can

spread. As the main differences between lightning and human caused fires are driven by the very fast-moving fires, when most of the burned area takes place. However, extreme as such might indeed be strongly expressed, so we softened the language to “more extreme” in figure 4 to reflect this better.

16. Line 446: I believe you should reference Tables S1 and S2 (not Tables S2 and S3).

Indeed, we now corrected this.

17. Table 1, last two rows: Pertaining to the strategy of bolding certain values, I understand how you can label fire weather as higher or lower risk, but not sure you can label biomass and forest cover as higher or lower risk. Same goes for Tables S1 and S2.

The rationale behind is that there tends to be a negative relation between fine fuel/cured grass load and biomass/tree cover. However, we agree that this is not universal and hence have removed this for both variables.

18. I’m also curious as to how much total area is burned by human and lightning fires and by ecoregion. I noted one of the figures in the Supplemental summarized by number of fires, but I’d also like to see area burned.

Some of this information was given in the introduction (L60-68). We have now expanded this sentence in the introduction to include burned area estimates per ecoregion:

“The dataset includes daily rate-of-spread for 214 individual fires including 2939 fire days (sum of the days with active fire detections for each fire) and covers 21,558 km² burned area, of which 13,699 km² are located in Northern California and 7,859 km² in the Mediterranean California ecoregion.”

Also:

“Lightning ignitions are the cause of 42.1% of the fires and 44.4% of the burned area in our database, while human caused fires represent 39.3% of fires and 33.7% of burned area (Figure S3). We note there is a residual set of fires that contribute to total fire number (18.6%) and burned area (21.9%) for which the cause is unknown”

19. Fig. S8: “ratio”, not “ration”.

Corrected

20. Table S2: forest cover looks like proportion and not percent.

Indeed, we have now converted the values to %.

21. Line 104-106: This sentence is confusing.

We agree that this sentence was not well formulated, more so because “fire” should have read “fire days”. We have now reformulated this sentence as follows: “Days where fires move slowly contribute relatively little to the overall burned area, so that the mean fire spread rate normalized

by burned area is higher, with a mean of 5.6 km day⁻¹ for human caused fires and 3.5 km day⁻¹ for lightning caused fires (Figure 2)."

22. Lines 111-112 (and elsewhere): You can delete "in our database".

Changed.

23. Line 128: "difference", not "change".

Changed.

24. I may have missed it, but how are "forests" and "shrubs/grass" differentiated? Describe in Methods.

These are based on the CALVEG vegetation map. This is mentioned in the methods section as: "To analyze the fire ecosystem impact and tree mortality we separate between forest and non-forest areas based on the CALVEG vegetation map for the year 2011 ³⁹."

25. Line 153: Probably the most notable way in which humans have changed fire regimes is through fire exclusion.

Agreed, we adapted the sentence to mention fire exclusion as follow: "One of most notable, besides fire exclusion, is the increase in ignition frequency, either directly by human agents or indirectly from damages to infrastructure such as a downed powerline during a wind storm."

26. Discussion: I'd like to see a paragraph or two devoted to how your findings compare and contrast to related studies that evaluated fire spread, fire severity/effects, and daily weather.

We now expand the discussion considerably to better frame our results in the context of previous research. We added the following paragraph: "The fire weather on the day of ignition is an important driver of rate-of-spread (Table 1, S1-2), with fire events occurring during extremes in fire weather more likely to escape early containment. This is in line with previous literature on meteorological drivers of fire spread ²⁵, but limited data exists on how these relationships shape day to day variation in spread rates. Surprisingly, we found that spread rates are strongly skewed, with most days showing slow rates-of-spread, and with rare instances of very fast spread rates (Figure 2). A similar distribution in spread rates has been observed over the boreal region ²⁶. This is in contrast to the representation of fire in most models, which overestimate spread rate under average fire weather conditions, probably due to an incomplete representation of the nonlinear and additive relationships between fire weather and fire spread ²⁶. The high variability in fire spread rates likely originate from complex non-linear interactions between fire weather, fuels, and topography, which are often not fully captured in modelling approaches ²⁵."

We also enhanced the discussion with regard to fire severity by adding the following sentences: "While it has been shown previously that fire weather and fuel drive fire severity ²⁸, our results show that these environmental drivers impact fire severity through their impact on fire spread rate and fire intensity."

Reviewers' Comments:

Reviewer #1:

Remarks to the Author:

I believe the modified manuscript was much improved based on the reviewer's comments, but I still have major concerns about this manuscript.

Because lightning is mostly accompanied by rain, it is thought that the speed of the initial fire spread is slower than that of human fires.

In the case of fires caused by humans, no matter how high-resolution and latest satellite data are used, it will be difficult to detect in the satellite data if the weather conditions do not support it and the spread does not spread significantly.

In my opinion, since only cases that can be detected by satellites were selected for the cases selected in this study, of course, only environments in which the weather conditions are favorable for wildfires were selected.

For example, isn't the reason the high wind speed in a human fire is that the fire spreads to a size observable from the satellite only when the wind speed is high?

To counter this, I think authors need to show anomaly that removes climate values as well as absolute values for wind speed.

For this reason, I request to present the calculated results by performing a linear regression according to each fire type. For example, one could calculate how sensitive the rate-of-spread is to various weather parameters (even aboveground biomass or forest cover) within a group for each fire type. In conclusion, if you believe that this study shows that natural fires (lightning) are at low risk because they are accompanied by some precipitation, but human ignition is more risky as they have a very rapid rate-of-spread in very dry weather conditions. We hope to reach a conclusion that is large.

As for the mortality, as studied by Rogers et al (<https://www.nature.com/articles/ngeo2352>), it is also necessary to clarify whether it is different depending on the characteristics of the tree. Siberian wildfires are rather fast because they only leave scars on tree stumps, so the speed is fast, but mortality is not thought to be large.

L44: Explain details about CalFIRE. What is it?

Figure 2: Please put error bars (95% confidence interval) for every bar.

I found several errors in your main and supplementary figures especially for bold.

Reviewer #2:

Remarks to the Author:

I think the authors have done a good job addressing my concerns and comments. No further comments from me.

Reviewer #3:

Remarks to the Author:

The authors have done a nice job addressing the comments from myself and other reviewers. Nice paper!

I have a few remaining comments that I'd like the authors to consider

1. Title: I'm not so sure about the phrase 'landscape position' in the title. I usually use this term in

relation to topography, like ridgetop vs. valley bottom, which is not addressed in this study. I guess this term is extremely vague and has no clear meaning. How about something like 'Human-ignited fires result in more extreme fire behavior and ecosystem impacts'?

2. Line 53: 'Fire intensity can be described by ...'. Is this correct? Fire intensity can be 'influenced by' or 'predicted by' these factors, but fire intensity itself is quantified numerically.

3. Lines 117-119: I don't understand the normalized values. Can this be explained differently or better. Same goes for lines 458-459 (figure 2).

4. Line 146: this is not a 'strong relationship' in my opinion. It is a 'statistically significant relationship' or something to this effect. I recommend toning down this statement.

5. Line 152: are you referring instead to Fig. S8? Be sure to check that all references to figures are correct. Also on Fig. S8, as mentioned in the last round of reviews, I's skip the exponential p-values spit out by R and go with something more reasonable.

Response to reviewer comments

We thank the reviewers for their additional comments and suggestions and give below a detailed response in blue.

REVIEWER COMMENTS

Reviewer #1 (Remarks to the Author):

I believe the modified manuscript was much improved based on the reviewer's comments, but I still have major concerns about this manuscript.

Because lightning is mostly accompanied by rain, it is thought that the speed of the initial fire spread is slower than that of human fires. In the case of fires caused by humans, no matter how high-resolution and latest satellite data are used, it will be difficult to detect in the satellite data if the weather conditions do not support it and the spread does not spread significantly. In my opinion, since only cases that can be detected by satellites were selected for the cases selected in this study, of course, only environments in which the weather conditions are favorable for wildfires were selected. For example, isn't the reason the high wind speed in a human fire is that the fire spreads to a size observable from the satellite only when the wind speed is high?

To study fire spread rates, we focused on larger, multi-day fire events that had sizes greater than 300 ha in the California Fire Resources and Protection (FRAP) database. This size threshold was necessary because it is impossible to compute spread rates for smaller fires. We state this clearly in the methods section and in the main text. We then considered all fires in this analysis, including all human and lightning ignited fires using information on this attribute from the FRAP database. Our satellite analysis was able to quantify fire spread rates for all fires meeting these criteria, irrespective of whether they had a lightning or human ignition origin. While it is true our analysis focuses on larger fires, we respectfully maintain there is not a bias in the way we are creating this dataset and there is no difference in the detection efficiency for lightning and human caused fires.

The fires represented in the fire spread database account for 88% of the total burned area so that most of the fire affected area during our study period is represented in our analysis. We describe the number of wildfires and the area represented by the fire observations in the main text lines 65-73:

“The dataset includes daily rate-of-spread for 214 individual fires including 2939 fire days (sum of the days with active fire detections for each fire) and covering 21,558 km² of burned area. As a function of ecoregion, the dataset covers 13,699 km² in Northern California and 7,859 km² in Mediterranean California. Fire sizes represented in the dataset range from 4 to 1660 km² (Figure S2). The dataset captures a wide range of daily rate-of-spread values, which vary from 0 to 22.6 km d⁻¹. The fires represented here are only fires that were not contained after the first day of fire spread, and hence only include a small fraction of the total number of reported fires (8.3%). Nevertheless, this set of large fires represent 88% of the total area burned in California during this period⁶.”

To counter this, I think authors need to show anomaly that removes climate values as well as absolute values for wind speed. For this reason, I request to present the calculated results by performing a linear regression according to each fire type. For example, one could calculate how sensitive the rate-of-spread is to various weather parameters (even aboveground biomass or forest cover) within a group for

each fire type. In conclusion, if you believe that this study shows that natural fires (lightning) are at low risk because they are accompanied by some precipitation, but human ignition is more risky as they have a very rapid rate-of-spread in very dry weather conditions. We hope to reach a conclusion that is large.

We do not understand very well which results are being referred to by the reviewer, and what such an analysis would show us. We don't expect to find structural differences in the relation between fire rate-of-spread and environmental variables for human caused and lightning caused fires because the physics driving their spread is the same.

While we are willing to perform additional analysis to clarify or strengthen our results, it is unclear how the analysis described would do this. We do report in Table 1 several differences in environmental conditions that occur for human and lightning-caused wildfires. Human-caused wildfires tend to occur when it is windier and when there are higher rates of potential evapotranspiration, which would cause dead fuels to be drier and combust more quickly.

In the previous round of review, we made many changes to the abstract and to the main text to make it clear that it is the environmental conditions at the time of ignition and during initial growth stages that set human and lightning-caused fires apart, and lead to diverging fire severity impacts. These statements include:

Abstract: "We observe that human-ignited fires coincide with extreme fire weather and ignition locations in more open ecosystems. These characteristics contribute to more explosive growth in the first few days following ignition for human-caused fires as compared to lightning-caused fires."

End of introduction: "It is often assumed that large wildfires caused by human activity are more devastating than lightning-caused fires, and here we hypothesize that this may occur because of an increased probability of human ignition during periods of extreme fire weather. To address this hypothesis, we ask the following four questions ..."

Results: "The observed differences between human- and lightning-caused fires are shaped by different environmental conditions at the time of ignition and during initial phases of fire expansion. Human-caused fires occur under significantly more extreme fire weather than lightning-caused fires. For example, human-caused fires are ignited during periods with higher potential evapotranspiration and on windier days (Table 1)."

Together, these statements show that we are not arguing for fundamentally different relationships between environmental conditions and rates of spread for human and lightning-caused fires. Rather, many human-caused fires are ignited under conditions of more extreme fire weather.

As for the mortality, as studied by Rogers et al (<https://www.nature.com/articles/ngeo2352>), it is also necessary to clarify whether it is different depending on the characteristics of the tree. Siberian wildfires are rather fast because they only leave scars on tree stumps, so the speed is fast, but mortality is not thought to be large.

Different tree species have indeed different strategies to cope with recurrent fires. However, the differences between Eurasia and North America seem to be based on evolutionary pathways leading to a dominance of fire embracers in North America and fire avoiders in Eurasia, which on their turn influence the fire regime. We agree that surface fires can move rapidly in Siberian forests when surface

fuels are dry, and tree mortality can be quite low. However, the black spruce species in North America promotes crown fires with their ladder fuels. The crown fires are more exposed to the wind and move more rapidly. The data reported in Rogers et al. shows than North American wildfires spread more rapidly than fires in Eurasia because of these differences (see Table 1 of Rogers et al., 2015).

Within California a mixture of tree species with different fire adaptive traits are present, so that it is hard to draw a parallel between the two continental evolutionary pathways in fire adaptive traits documented for boreal forests and the mixed Californian forests. We also note that the study of Rogers et al. (2015) could not link fire behavior with rate-of-spread on a daily time scales, because the algorithms and data needed to compute rate of spread accurately had not been developed at the time. In this context, we note that our use of VIIRS satellite observations and the alpha hull algorithm to estimate rate-of-spread is a major advance that sets our paper apart from previous work.

We do separate our analysis between known ecosystem level differences in fire regimes in California, specifically between Mediterranean California and Northern California. This is described in the methods section as follow: “As California ecosystems are highly diverse, we performed our analysis separately for two large ecosystem groups with extensive fire occurrence in California, namely the level 2 ecoregion Mediterranean California and Northern California”.

L44: Explain details about CalFIRE. What is it?

We have now included the full name of CalFire and indicate that this is the abbreviation for the “California Department of Forestry and Fire Protection.” We added the following text to the first paragraph of the methods to address this reviewer concern:

“The FRAP database is maintained by the State of California and updated annually. The effort, led by CalFIRE, integrates information on fire perimeters from the US Forest Service, the Bureau of Land Management, and the National Park Service. Additional fire perimeters are entrained from other US federal agencies when available (e.g., the Dept. of Defense and Bureau of Indian Affairs) ⁶. The fire perimeters are digitized from multiple information sources including ground or aircraft global positioning system (GPS) coordinates, infrared imagery, photo interpretation, and mixed data sources. Key attributes associated with each fire include the start date, containment date, and the cause of ignition.”

Figure 2: Please put error bars (95% confidence interval) for every bar.

Figure 2 shows the probability distribution function for rate-of-spread within our database for the two different fire types, and as such we feel that error bars are not appropriate for this figure. The reviewer does raise a good point that in some way we need to statistically test whether the distributions are significantly different.

To address this reviewer suggestion, we now use the Kolmogorov-Smirnov test which indeed confirms that the distributions in rate-of-spread between human and lightning caused are indeed significantly different.

We have added the following sentence to the figure 2 caption: “The frequency distributions of daily rate-of-spread between human and lightning-caused fires as a function of fire day or burned area are significantly different ($p < 0.001$).”.

We also added the following sentence to the methods section: “We tested whether the distributions in daily fire rate-of-spread are significantly different between human and lightning caused fires using the Kolmogorov-Smirnov test.”

We also added the following sentence to the results section: “The distributions in fire-spread-rate are significantly different between human and lightning caused fires ($p < 0.001$) ...” and “As a function of burned area, the frequency distributions of the two fire types are also significantly different ($p < 0.001$).”

I found several errors in your main and supplementary figures especially for bold.

We have carefully reviewed the figures and tables and modified the caption in figure S6, S7 and S8. However, for Tables 1, S1 & S2, we indeed only indicate meteorological values in bold, not vegetation related variables, which was indeed not specified in the caption. We now added to the captions that this is only for meteorological variables.

Reviewer #2 (Remarks to the Author):

I think the authors have done a good job addressing my concerns and comments. No further comments from me.

We appreciate that reviewer 2 was satisfied by how we addressed their comments and suggestions.

Reviewer #3 (Remarks to the Author):

The authors have done a nice job addressing the comments from myself and other reviewers. Nice paper!

I have a few remaining comments that I'd like the authors to consider.

1. Title: I'm not so sure about the phrase 'landscape position' in the title. I usually use this term in relation to topography, like ridgetop vs. valley bottom, which is not addressed in this study. I guess this term is extremely vague and has no clear meaning. How about something like 'Human-ignited fires result in more extreme fire behavior and ecosystem impacts'?

We agree and have now changed the title to the one suggested here by the reviewer.

2. Line 53: 'Fire intensity can be described by ...'. Is this correct? Fire intensity can be 'influenced by' or 'predicted by' these factors, but fire intensity itself is quantified numerically.

We agree that “described” is not the best term, we have changed this to “estimated”. We also added fuel “consumed” to this sentence, as it is of course not directly the fuel load, but the fuel consumed which determines the fire intensity.

3. Lines 117-119: I don't understand the normalized values. Can this be explained differently or better. Same goes for lines 458-459 (figure 2).

We present both mean values, through time and across burned area, to take into account the fact that on days where the fire doesn't spread fast not much area is burned either. A simple example for a fire burning 2 days might clarify this. Imagine that on day one the fire spreads at 1km/day and burns 2km², and the second day at 10km/day and burns 20km². If we take the mean through time the mean fire spread rate is 5.5km/day. However, when normalizing as a function of burned area, we obtain 9.18km/day spread rate ((1*2+10*20)/22), as almost the entire area burned under 10km/day fire spread rate.

We have now added to the methods section the following text and formula:

"The mean in fire spread rate (Figure 2) was assessed both with and without normalization by burned area. To normalize by burned area, we applied the following equation:

$$\frac{\sum_i^n BA_i * RoS_i}{\sum_i^n BA_i} \quad (1),$$

where BA_i is the burned area on day i , RoS_i is the fire spread rate at day i , and n is the total fire days considered."

We have also added a reference to the methods section in the main text.

4. Line 146: this is not a 'strong relationship' in my opinion. It is a 'statistically significant relationship' or something to this effect. I recommend toning down this statement.

We have now rephrased this to "statistically significant relationship", as suggested by the reviewer.

5. Line 152: are you referring instead to Fig. S8? Be sure to check that all references to figures are correct.

Indeed, we now correct the figure reference here to Figure S8. We further checked all figure references and corrected where necessary to be sure that all are correct now.

Also on Fig. S8, as mentioned in the last round of reviews, I's skip the exponential p-values spit out by R and go with something more reasonable.

We agree with the reviewer, during the previous revision we indeed adjusted the exponential p-values to the equally informative and more reasonable <0.001. The exponential p-values in S8 slipped through the previous revision. We have adapted these now as well.

Reviewers' Comments:

Reviewer #1:

Remarks to the Author:

I think the manuscript is improved to accept Nature Communications. One thing I would like to suggest is that stronger wind in human-caused ignition might be related to early control by firefighting. I think weaker wind cases can be easily controlled by firefighting, so comparing results between human- and lightning-caused events tend to be significant. I was wondering if the authors have any evidence and data for successful firefighting cases, it is better to mention them in the discussion part.

Response to reviewer comments

We thank the reviewers for their additional comments and suggestions and give below a detailed response in blue.

Reviewer #1 (Remarks to the Author):

I think the manuscript is improved to accept Nature Communications. One thing I would like to suggest is that stronger wind in human-caused ignition might be related to early control by firefighting. I think weaker wind cases can be easily controlled by firefighting, so comparing results between human- and lightning-caused events tend to be significant. I was wondering if the authors have any evidence and data for successful firefighting cases, it is better to mention them in the discussion part.

We agree with the reviewer that having detailed information on firefighting resources allocated to each fire would allow for a more detailed analysis. Unfortunately, firefighting resources applied to each fire are currently not readily available. We have now added a sentence to the discussion section to point out the importance of firefighting resource availability: "An important next step should be to determine the degree to which fire suppression resource allocation differs for human and lightning-caused fires and how this propagates to fire growth and fire damage."

Furthermore, while recalculating some statistics we found a small bug in our code to produce figure 3. We have now regenerated figure 3 with the corrected code. While some changes are apparent, these do not qualitatively influence the results or the interpretation of those.